# The Regulatory Cross-Talk between microRNAs and Novel Members of the B7 Family in Human Diseases: A Scoping Review

**DOI:** 10.3390/ijms22052652

**Published:** 2021-03-06

**Authors:** Noora Karim Ahangar, Nima Hemmat, Mohammad Khalaj-Kondori, Mahdi Abdoli Shadbad, Hani Sabaie, Ahad Mokhtarzadeh, Nazila Alizadeh, Afshin Derakhshani, Amir Baghbanzadeh, Katayoun Dolatkhah, Nicola Silvestris, Behzad Baradaran

**Affiliations:** 1Department of Animal Biology, Faculty of Natural Sciences, University of Tabriz, Tabriz 5166616471, Iran; noora.k97@ms.tabrizu.ac.ir (N.K.A.); khalaj49@gmail.com (M.K.-K.); 2Immunology Research Center, Tabriz University of Medical Sciences, Tabriz 5165665811, Iran; nima.hemmat1995@gmail.com (N.H.); abdoli.med99@gmail.com (M.A.S.); ahad.mokhtarzadeh@gmail.com (A.M.); nazilaalizadeh08@gmail.com (N.A.); afshin.derakhshani94@gmail.com (A.D.); amirbaghbanzadeh@gmail.com (A.B.); dolatkhah.k45@gmail.com (K.D.); 3Student Research Committee, Tabriz University of Medical Sciences, Tabriz 5166614766, Iran; sabaieh@tbzmed.ac.ir; 4IRCCS IstitutoTumori “Giovanni Paolo II” of Bari, 70124 Bari, Italy; 5Department of Biomedical Sciences and Human Oncology, University of Bari “Aldo Moro”, 70124 Bari, Italy

**Keywords:** B7, microRNA, human diseases, cancer, immune checkpoint, immunotherapy, gene therapy

## Abstract

The members of the B7 family, as immune checkpoint molecules, can substantially regulate immune responses. Since microRNAs (miRs) can regulate gene expression post-transcriptionally, we conducted a scoping review to summarize and discuss the regulatory cross-talk between miRs and new B7 family immune checkpoint molecules, i.e., B7-H3, B7-H4, B7-H5, butyrophilin like 2 (BTNL2), B7-H6, B7-H7, and immunoglobulin like domain containing receptor 2 (ILDR2). The current study was performed using a six-stage methodology structure and Preferred Reporting Items for Systematic Reviews and Meta-Analyses (PRISMA) guideline. PubMed, Embase, Scopus, Cochrane, ProQuest, and Google Scholar were systematically searched to obtain the relevant records to 5 November 2020. Two authors independently reviewed the obtained records and extracted the desired data. After quantitative and qualitative analyses, we used bioinformatics approaches to extend our knowledge about the regulatory cross-talk between miRs and the abovementioned B7 family members. Twenty-seven articles were identified that fulfilled the inclusion criteria. Studies with different designs reported gene–miR regulatory axes in various cancer and non-cancer diseases. The regulatory cross-talk between the aforementioned B7 family molecules and miRs might provide valuable insights into the pathogenesis of various human diseases.

## 1. Introduction

Immune checkpoints can considerably regulate immune responses [1]. These molecules are critical for maintaining self-tolerance and preventing the stimulation of immune responses against normal peripheral tissues. Indeed, suppressing inhibitory axes, e.g., the immune checkpoint axis of cytotoxic T lymphocyte antigen 4 (CTLA-4) and programmed death-ligand 1 (PD-L1), has revolutionized cancer immunotherapy [2].

The B7 family is a group of immune checkpoints commonly expressed in different immune cells, such as antigen-presenting cells, T cells, B cells, natural killer cells, and various tissues. They play a crucial role in immune response; for example, they have substantial roles in directing the fate of T cells by binding their receptors. Various members of the B7 family have been identified, e.g., *B7.1* (*CD80*), *B7.2* (*CD86*), *B7-H1* (*PD-L1*, or *CD274*), *B7-DC* (*PD-L2*, *PDC1LG2*, or *CD273*), *B7-H2* (*ICOSL*: inducible T-cell co-stimulator ligand, or *CD275*), *B7-H3* (*CD276*), *B7-H4* (*VTCN1*), *B7-H5* (*VISTA*: V-domain Ig suppressor of T cell activation, *Dies1*: differentiation of embryonic stem cells 1, or *C10orf54*), butyrophilin like 2 (*BTNL2*), *B7-H6* (*NCR3LG1*: natural killer cell cytotoxicity receptor 3 ligand 1), *B7-H7* (*HHLA2*: human endogenous retro virus–H long repeat-associating 2), and immunoglobulin like domain containing receptor 2 (*ILDR2*) [2,3,4,5,6]. Indeed, BTNL2 and ILDR2 are introduced as B7-like molecules, and further investigation is needed. B7 family genes have been linked with various pathological conditions, e.g., cancers, infections, autoimmune diseases, and transplantation complications [6]. Thus, a better understanding of their biology might pave the way for introducing novel strategies to treat the abovementioned diseases and complications.

MicroRNAs (miRs), as small, non-coding RNAs, can bind to their complementary sequences, which are often the mRNA 3′-untranslated regions (3′UTR) of their targets. Since miRs can cleave their target mRNAs by guiding the RNA-induced silencing complex (RISC) to target mRNAs, in order to direct the cleavage of mRNA through Argonaute (AGO) endonuclease activity [7], destabilize their target mRNAs via cutting their poly(A) tail, and make the translation of their target mRNA less effective, they are considered potent post-transcriptional gene regulators [8,9,10]. In higher eukaryotes, miRs can regulate the expression of approximately 60% of genes. It is well-established that miRs can contribute to many biological processes, e.g., cell growth, differentiation, metabolism, and immune response regulation [11,12]. Indeed, miRs can modulate the function of immune cells and regulate the expression of immune checkpoints [13,14,15]. Thus, alteration in miR expression is involved in the pathogenesis of various human diseases, like cancers [11,13,16]. Moreover, miRs can regulate the expression of B7 family members in various diseases; thus, there is a need to properly understand the scope and effect of this regulation in human diseases [17,18].

This scoping review focuses on novel B7 family members, i.e., *B7-H3*, *B7-H4*, *B7-H5*, *BTNL2*, *B7-H6*, *B7-H7*, and *ILDR2*. This study aimed to map the current regulatory cross-talk between the above-mentioned members of the B7 family and miRs. Moreover, we used bioinformatics approaches to identify other potential miRs targeting these genes. Since miRs are promising targets for therapeutic applications [19,20], the results of this review might provide valuable insights for a better understanding the regulatory cross-talk between novel B7 immune checkpoint members and miRs in the development of various human diseases, as well as provide new molecular targets for use in therapeutic and clinical applications.

## 2. Methods

The methodology for the current scoping review was according to the framework recommended by Arksey and O’Malley [21] and later enhanced by Levac et al. [22]. It consists of five distinctive phases: (1) identifying the research question; (2) identifying relevant studies; (3) study selection; (4) charting the data; and (5) collating, summarizing, and reporting results. An optional sixth phase, consultation, was not a part of our scoping review. This study was also carefully guided by the Preferred Reporting Items for Systematic Reviews and Meta-Analyses Extension for Scoping Reviews (PRISMA-ScR) Checklist [23].

### 2.1. Identifying the Research Question

This study aimed to map the current literature on new B7 family members and miRs’ regulatory axis. To address this aim, we sought to answer the upcoming question: precisely what is known from existing literature about the regulation of new B7 family molecules in human diseases by miRs?

### 2.2. Identifying Relevant Studies

PubMed, Embase, Scopus, Cochrane, ProQuest, and Google Scholar were searched based on the specific search tips of each database without any restriction, in order to recognize all potentially eligible publications. MeSH and Emtree terms were used if available. The search strategy for PubMed and Embase databases are described in Appendix A. The last search was conducted on 5 November 2020. We also included articles identified from hand-searching or reference lists of the selected studies.

### 2.3. Selecting Studies

The included studies fulfilled the following criteria: (1) explicitly discussing the regulatory axes between the new B7 family members and miRs, (2) evaluating human specimens or cell lines, and (3) written in English. The exclusion criteria were (1) studies on animals and (2) studies that were not primary/original research. First, publication titles and abstracts were independently screened by two investigators (N.K.A. and H.S.) for eligibility, according to the above-mentioned criteria. In the second phase, the full-text assessment of the selected studies was conducted, and studies that met the eligibility criteria were included in the final data analysis. Any disagreements were solved by involving a third reviewer (B.B.), if required.

### 2.4. Charting the Data

Two authors (N.K.A. and H.S.) independently extracted data into a predefined charting form in Microsoft Excel. It provided information about regulatory axes, samples, methods, diseases, and key findings of each study.

### 2.5. Collating, Summarizing, and Reporting the Results

Both quantitative and qualitative analyses were carried out. We presented a descriptive numerical summary of the chosen studies’ features for the quantitative portion. We prepared a narrative review, addressing our abovementioned research question for the qualitative assessment, considering the importance of findings in the broader context proposed by Levac et al. [22].

### 2.6. Bioinformatics Analysis

The miRDB database (accessed on 15 February 2021) [24] was used to predict other possible new B7 family members and miRs’ interaction. Only functional human miRs with a target score >80 were included. According to miRDB, a predicted target with a score >80 is most likely to be real. Also, miRs with >2000 predicted targets in the genome were excluded. The Kyoto Encyclopedia of Genes and Genomes (KEGG) pathway enrichment analysis was performed by miRPathDB v2.0 to obtain insights on the pathways that are targeted by predicted miRs [25]. Specifically, the focus was on the “Immune system” in the KEGG database. Selection of data based on strong experimental evidence and with a minimum of two significant miRs per pathway was used as criteria for the miRPathDB query.

## 3. Results

The flowchart of literature identification, inclusion, and exclusion is demonstrated in Figure 1.

A total of 199 articles were identified through database searching and hand-searching, of which 96 were duplicates. Seventy-four articles were excluded during the title and abstract review, based on irrelevance to the inclusion criteria. The full-text assessment of the remaining 29 records was conducted, and two more records were excluded because they were not human studies [26,27]. Finally, a total of 27 eligible papers remained [28,29,30,31,32,33,34,35,36,37,38,39,40,41,42,43,44,45,46,47,48,49,50,51,52,53,54]. The characteristics of the included studies are summarized in Table 1.

The included articles were published between 2009 and 2020. Twenty studies were performed in China [29,30,31,33,34,38,39,40,42,43,44,45,46,47,49,50,51,52,53,54], five in the United States [28,32,37,41,48], one in Canada [36], and one in Norway [35]. Clinical and experimental data from different human specimens, i.e., tissue, blood, serum, bone marrow, nasopharyngeal secretion, skin, and human cell lines, were evaluated. Different approaches like bioinformatics analysis were utilized to predict interactions between miRs and the desired genes. Furthermore, various molecular techniques and cell-based assays, e.g., real-time PCR, Western blotting, protein lysate microarray, miR array, gene silencing techniques, sequencing, cell viability assays, cell proliferation assays, cytotoxicity assays, and cell death assays were used to assess these interactions. Luciferase activity assay and transfection technique were also used to validate the regulatory mechanisms.

### 3.1. B7-H3

Twenty-one papers studied *B7-H3* regulatory axes [28,29,30,31,32,33,34,35,37,39,41,42,43,44,45,48,50,51,52,53,54]. Thirty-three miRs involved in the regulatory axes and diseases pathogenesis, including miR-29 (in Burkitt lymphoma, adenocarcinoma, brain tumor, hepatoblastoma, neuroblastoma, sarcomas, Wilms’ tumor, cutaneous melanoma, breast cancer, central nervous system (CNS) neuroblastoma, allergic asthma, diffuse brain glioma, colorectal cancer, medulloblastoma, *Mycoplasma pneumoniae* pneumonia), miR-539 (in glioma), miR-124 (in osteosarcoma), miR-187 (in psoriasis, colorectal cancer, and clear-cell renal cell carcinoma), miR-380–5p, miR-125b-2, miR-363, miR-940, miR-214, miR-34b, miR-665, miR-593, miR-555, miR-885–3p, miR-567, miR-297, miR-187–3p, miR-124a-1, miR-326, miR-601, miR-506 and miR-708 (in breast cancer), miR-155, miR-143, miR-192, miR-378, miR-1301–3p, miR-335–5p and miR-28–5p (in colorectal cancer), miR-506 (in mantle cell lymphoma), miR-214 (in multiple myeloma), miR-1253 (in medulloblastoma), miR-145 (in lung cancer and colorectal cancer), and miR-199a (in cervical cancer).

### 3.2. B7-H4

Three studies investigated the regulatory axes between *B7-H4* and miRs in human diseases [38,46,53]. The studied miRs and diseases were as follows: miR-155, miR-143, miR-1207 (in colorectal cancer), hsa-miR-299–5p, hsa-miR-2115–3p, hsa-miR-1284, hsa-miR-1265, hsa-miR-3178, hsa-miR-204–3p, hsa-miR-3646, hsa-miR-3686, hsa-miR-4279, hsa-miR-302e, hsa-miR-138–1-3p, hsa-miR-31–5p, hsa-miR-4290, hsa-miR-4306, hsa-miR-483–3p, hsa-miR-744–5p, hsa-miR-183–3p, hsa-miR-24–1-5p, hsa-miR-4324, hsa-miR-519e-3p, hsa-miR-3202, hsa-miRPlus-G1246–3p, hsa-miR-361–5p, hsa-miR-2116–5p, hsa-miR-335–3p, hsa-miR-4258, hsa-miR-4284, hsa-miR-3685, hsa-miR-302a-3p, hsa-miR-361–3p, hsa-miR-33a-5p, hsa-miR-642a-5p, hsa-miR-33b-5p, hsa-miR-642b-5p, hsa-miR-3651, hsa-miR-7–5p, hsa-miR-1260a, hsa-miR-21–5p, hsa-miR-149–3p, hsa-miR-130b-3p, hsa-miR-525–5p, hsa-miR-4288, hsa-miR-513a-5p, hsa-let-7f-5p, hsa-miR-937, hsa-miR-31–3p, hsa-miR-1290, hsa-miR-196a-5p, hsa-miR-1246, hsa-miR-374c-5p, hsa-miR-615–3p, hsa-miR-27b-3p, hsa-miR-1973, hsa-miR-17–3p, hsa-miR-186–5p, hsa-miR-3676–3p, hsa-let-7c, hsa-miR-335–5p, hsa-miR-122–3p, hsa-miR-148a-3p, hsa-miR-520d-5p, and hsa-miR-21–3p (in pancreatic cancer).

### 3.3. B7-H5

Two studies investigated the miR-16/*B7-H5* and miR-125a-5p/*B7-H5* regulatory axes in active Crohn’s disease and gastric cancer, respectively [36,47].

### 3.4. B7-H6

One study reported that the miR-93/*B7-H6*, miR-195/*B7-H6*, and miR-340/*B7-H6* regulatory axes might contribute to regulating immune evasion in breast cancer [49].

### 3.5. B7-H7

One study indicated that the regulatory axes of hsa-miR-3116/*B7-H7* and hsa-miR-6870–5p/*B7-H7* contribute to the regulatory mechanisms of human tumors [40].

### 3.6. ILDR2

There were no studies investigating the interaction of miR with this gene.

### 3.7. BTNL2

There were no papers investigating the interaction of miR with this gene.

### 3.8. Bioinformatics Analysis

Predictions of possible gene–miR interactions and KEGG pathway enrichment analysis revealed that four miRs are annotated for KEGG pathways in the category immune system (Figure 2): hsa-miR-29b-3p was annotated for “Chemokine signaling pathway”, “T cell receptor signaling pathway”, and “Leukocyte transendothelial migration”; hsa-miR-29a-3p was annotated for “T cell receptor signaling pathway” and “Complement and coagulation cascades”; hsa-miR-125a-5p was annotated for “T cell receptor signaling pathway” and “Chemokine signaling pathway”; and hsa-miR-486-5p was annotated for “Leukocyte transendothelial migration”.

To further understand and visualize the interactions, a gene–miR pathway network was constructed (Figure 3). 

Based the interactions, we identified potential new interactions between the B7 family members and miRs: (1) hsa-miR-29b-3p/*B7-H3*, (2) hsa-miR-29a-3p/*B7-H3*, (3) hsa-miR-125a-5p/*B7-H4*, and (4) hsa-miR-486-5p/*B7-H4*. The association of different isoforms of miR-29 with *B7-H3* was supported by the reviewed literature.

## 4. Discussion

### 4.1. B7-H3

*B7–H3*, also referred to as *CD276*, can regulate the stimulation and inhibition of T cells [55,56]. A variety of cells, e.g., natural killer cells, activated T-cells, dendritic cells, macrophages, and non-hematopoietic cells, can express *B7-H3* [57]. Preliminary findings reported that B7-H3 could promote CD4+ and CD8+ T cell proliferation by T cell receptor (TCR) stimulation using immobilized Ig fusion protein [55]. However, it is well-established that B7–H3 can suppress the activation of CD4+ T-cell and the release of effector cytokines [58,59]. This suppression might facilitate the function of transcription factors like nuclear factor of activated T cells (NF-AT), nuclear factor kappa B (NF-κB), and activator protein 1 (AP-1), playing significant roles in T cell function [58]. Moreover, *B7-H3* overexpression has been identified in various cancers, e.g., breast [60], lung [61], kidney, prostate [62], and ovarian cancer [63]. Furthermore, the inhibition of *B7-H3* has decreased angiogenesis in medulloblastoma, indicating its essential role in tumor angiogenesis [37]. As an overexpressed oncogene in various cancers, MYC has a critical role in cancer development, e.g., angiogenesis, apoptosis, proliferation [64,65]. Since MYC inhibition has been associated with the suppressed expression of *B7-H3* in medulloblastoma cells, the MYC-*B7-H3* regulatory axis can play an essential role in regulating angiogenesis [37]. It has been indicated that *B7-H3* knockdown can repress the PI3K/Akt pathway, resulting in decreased STAT3 activity. Since STAT3 can promote the expression of matrix metalloproteinase 2 (*MMP2*) and matrix metalloproteinase 9 (*MMP9*), B7-H3 can regulate the expression of *MMP2* and *MMP9* [66]. Moreover, *B7-H3* can be involved in inflammatory conditions, e.g., sepsis and bacterial meningitis [51]. Since the mRNA expression of *B7–H3* is not as remarkable as its protein expression, the post-transcriptional regulating process might have a considerable effect [57].

The expression of *B7-H3* is controlled by various factors, e.g., cytokines, oncogenes, long non-coding RNAs (lncRNAs), and miRs [30,34,37,53]. Although several miRs can directly target the mRNA of *B7-H3*, several miRs can indirectly regulate *B7-H3* expression by targeting an intermediate mRNA [13]. It has been indicated that isoforms of mir-29 can inversely regulate *B7-H3* expression in various cancers, e.g., neuroblastoma [28], cutaneous melanoma [41], breast cancer [35], diffuse brain glioma [29], colorectal cancer [44], and medulloblastoma [37]. Growing evidence highlights the crucial role of dysregulated miR-29c in tumor development and the prognosis of affected patients [67,68,69]. The low expression of miR-29c has been associated with *TP53* mutation in breast cancer patients. Indeed, the tumors with low miR-29c expression lack TP53 tumor suppressor activity [35]. Of interest, miR-29 can repress STAT3, integrin, vascular endothelial growth factor (VEGF), NF-κB, and the PI3K/AKT signaling pathway, and upregulate the expression of phosphatase and tensin homolog (*PTEN*) and *p53* [37]. A study has found that 13 miRs, i.e., miR-363, miR-326, miR-214, miR-29c, miR-665, miR-940, miR-708, miR-601, miR-34b, miR-380–5p, miR-124a, miR-593, and miR-885–3p, can directly target *B7-H3*. Indeed, miR-29c overexpression has been indicative of the improved prognosis of breast cancer patients [35]. This correlation between miR-29c and *B7-H3* has also been identified in other diseases. For instance, there is a negative association between miR-29c expression and *B7-H3* expression in children with asthma deterioration. Indeed, miR-29c can target *B7-H3* in macrophages and alter the differentiation of T helper cells [51]. One study has demonstrated a negative correlation between miR-29c and *B7-H3* in children with *Mycoplasma pneumoniae* pneumonia [33]. Moreover, Astragaloside IV, which can suppress cell proliferation in colorectal cancer, can downregulate *B7-H3* via miR-29c and enhance the chemosensitivity of tumor cells [44]. In line with this, miR-29c can enhance the sensitivity to cisplatin in nasopharyngeal carcinoma cells [70]. In addition, miR-29c can improve the chemosensitivity of non-small-cell lung cancer cells to cisplatin by regulating the PI3K/Akt pathway [71].

Furthermore, the downregulation of miR-335–5p, miR-1301–3p, and miR-28–5p can upregulate *B7-H3* expression in advanced colorectal cancer [43]. Besides, there is a negative association between *B7-H3* and miR-187 in clear-cell renal cell carcinoma and colorectal cancer [52]. Also, miR-187, which is downregulated in cytokine-stimulated keratinocytes and psoriasis, can suppress keratinocyte hyperproliferation via negatively regulating *B7-H3* expression [39]. In mantle cell lymphoma, miR-506 downregulation and *B7-H3* overexpression have been associated with increased cell proliferation and tumor migration [54]. The altered expression of miR-506 in cancers has resulted from the methylation of the promoter and modifications in upstream transcription factors [72]. In glioma, miR-539 upregulation, which can suppress *B7-H3* expression, can repress cancer development by targeting caspase recruitment domain-containing, membrane-associated, guanylate kinase protein-1 (CARMA1), sperm-associated antigen 5 (SPAG5), and the gene encoding matrix metalloproteinase 8 (MMP8) [34]. In addition, the upregulation of miR-214 can result in the proliferation and migration of tumor cells in ovarian and gastric cancers [73]. In multiple myeloma, lncRNA NEAT1, by targeting miR-214, upregulates *B7-H3*, promoting M2 macrophage polarization and tumor development. Indeed, miR-214 and *B7-H3* can serve as valuable prognostic factors and be used as promising targets for treating multiple myeloma patients [30].

Transforming growth factor-beta 1 (TGF-β1) can upregulate the *B7-H3* expression and pave the way for immune evasion of colorectal cancer cells via the miR-155/miR-143 axis. Indeed, miR-143 can inhibit *B7-H3* expression in colorectal cancer cells [53]. The downregulation of miR-124 can increase the expression of *B7-H3* in osteosarcoma. It has been reported that miR-124 might decrease proliferating cell nuclear antigen (PCNA) and cyclin D1, and elevate apoptotic protein poly-ADP ribose polymerase (PARP). Also, miR-124 mimics might increase the therapeutic efficacy of monoclonal antibodies targeting B7-H3 [42]. Furthermore, miR-1253 can inhibit cell proliferation and promotes the apoptosis of tumor cells in medulloblastoma. Indeed, miR-1253 restoration and *B7-H3* silencing can substantially decrease the migration of medulloblastoma cells [32]. Zhou et al. found that the low expression of miR-192, miR-378, and miR-145 can result in *B7-H3* overexpression and immune evasion in colorectal cancer [53]. Consistent with this, the downregulation of miR-145 has been associated with increased *B7-H3* expression, facilitating lymph node metastasis in lung cancer patients [31]. In cervical cancer, miR-199a, which targets *B7-H3*, can inhibit cell proliferation, invasion, and migration [50].

### 4.2. B7-H4

*B7-H4*, also known as *B7x*, *B7S1*, and *VTCN1*, can inhibit cytokine production, proliferation, cell cycle progression, and the stimulation of CD4^+^ and CD8^+^ T cells [74,75]. Although its transcripts can be found in various tissues, its protein has low expression in most human normal tissues [76]. *B7-H4* expression is positively correlated with cancer development in patients with gastric cancer [77], glioma [78], squamous cell esophageal carcinoma [79], renal cell carcinoma [80], pancreatic cancer [81], cholangiocarcinoma [18], ovarian cancer [82], and lung cancer [83]. Since *B7-H4* has been associated with cancer development, it can be an appealing target for treating cancer patients [76].

It has been shown that miR-125b-5p has an anti-inflammatory role and can regulate interleukin (IL)-1β-induced inflammatory genes by targeting the TNF receptor associated factor (TRAF6)/mitogen-activated protein kinase (MAPK)/NF-κB pathway in human osteoarthritic chondrocytes [84]. However, miR-125b-5p overexpression in macrophages can increase IL-2 secretion and the proliferation of CD8^+^ T cells. Indeed, miR-125b-5p can target *B7-H4* and facilitate inflammation [85]. In line with this, *B7-H4* overexpression has been associated with poor prognosis in colorectal cancer patients [86]. In 24.4% of colorectal cancer patients, single-nucleotide polymorphism (SNP) rs13505 GG of *B7-H4* can confer an alternate binding site for miR-1207–5p, which might result in downregulation of this gene [46]. Furthermore, TGF-β1 can upregulate *B7-H4* and facilitate immune escape via the miR-155/miR-143 axis in colorectal cancer [53].

In 2017, 62 hsa-miRs were identified as regulating *B7-H4* in pancreatic cancer [38]. These miRs were mentioned above in the Results section.

### 4.3. B7-H5

B7-H5, also known as VISTA, C10orf54, Dies1, and PD-1H, is a type-I membrane protein that can stimulate terminal differentiation of embryonic stem cells (ESCs) into cardiomyocytes/neurons via the bone morphogenetic protein (BMP) signaling pathway [87]. It has been reported that miR-125a-5p can directly repress the transcription of *B7-H5* and inhibit ESC differentiation [26]. B7-H5 also plays a pivotal regulatory function in adipocyte differentiation independently from BMP signaling. In particular, the elevated level of *B7-H5* has been shown exclusively in differentiated fat cells and blocked adipocyte differentiation [88]. In *B7-H5* knockout mice, the elevation of inflammatory cytokines can result in chronic multi-organ inflammation, indicating the critical role of B7-H5 in suppressing inflammation [89]. In Crohn’s disease, there is a negative association between *B7-H5* expression and hsa-miR-16–1 [47]. B7-H5 can serve as a ligand and receptor on T cells, suppressing the activation of naïve and memory T cells [90,91]. The presence of two PKC binding sites in the cytoplasmic region of B7-H5 might indicate that B7-H5 is a receptor [92,93]. *B7-H5* can be overexpressed in cancer-associated/cancer-adjacent gastric myofibroblasts. However, *B7-H5* expression is generally downregulated in epithelial gastric cancer cells. This can be explained by *B7-H5* promoter methylation, the overexpression of miR-125a-5p, or a combination of both, and even the existence of mutant *p53* [36]. Indeed, the downregulation of *B7-H5* has been associated with de-differentiation and triggered epithelial–mesenchymal transition (EMT) in epithelial cells.

### 4.4. B7-H6

B7-H6, also known as NCR3LG1, is a ligand for the NKp30 [94]. B7-H6 sequence is functionally similar to the other B7 family members. Although *B7-H6* is not found in normal human tissues, it is highly expressed in cancers, e.g., renal cell carcinoma, leukemia, breast cancer, ovarian cancer, and sarcomas [95]. Various factors can regulate *B7-H6* expressions, e.g., protease inhibitors, proinflammatory cytokines, natural killer cells, and miRs. Histone deacetylase inhibitors (HDACi) and metalloprotease inhibitors can regulate the *B7-H6* expression at transcription and post-transcriptional levels, respectively [96]. Following stimulation of CD14^+^CD16^+^ neutrophils and monocytes, B7-H6 can be expressed on these proinflammatory immune cells [97]. Tumoral B7-H6 can be recognized and eliminated via natural killer cells. However, metalloproteases can cleave B7-H6 and shield tumor cells from natural killer-mediated immune responses [98]. Bioinformatics analysis has predicted that miR-93, miR-195, and miR-340 can regulate immune responses by targeting *B7-H6* in breast cancer cells [49].

### 4.5. B7-H7

*B7-H7*, which has previously been referred to as *B7-H5*, is known as the human endogenous retro virus–H long repeat-associating 2 (*HHLA2*) [99,100]. Its receptors can be found on various immune cells, e.g., monocytes, T cells, B cells, and dendritic cells. TMIGD2, which is referred to as CD28 homolog, is one of the identified B7-H7 receptors [101]. In antigen-presenting cells, B7-H7 co-stimulates the proliferation of naïve T cell and cytokine production across TMIGD2 by serine–threonine kinase AKT phosphorylation. However, the second B7-H7 receptor on activated T cells can exert a coinhibitory role, because activated T cells do not express TMIGD2. The identification of the second receptor might clarify the role of B7-H7 in T cell activation and the tumor microenvironment [102]. It has been reported that *B7-H7* is upregulated in lung cancer, osteosarcoma, and breast cancer, and its elevated expression is correlated with a poor prognosis in affected patients [103]. BATF in B lymphocytes and SMAD in monocytes might be involved in the dysregulation of *B7-H7* in kidney clear-cell carcinoma. It has been indicated that hsa-miR-6870–5p and hsa-miR-3116 might have a role in this modulatory mechanism [40].

### 4.6. Bioinformatics Analysis

Based on our results, four potential new interactions between B7 family members and miRs have been identified: (1) the hsa-miR-29b-3p/*B7-H3* axis, (2) the hsa-miR-29a-3p/*B7-H3* axis, (3) the hsa-miR-125a-5p/*B7-H4* axis, and (4) the hsa-miR-486-5p/*B7-H4* axis. Of these four interactions, the association of different isoforms of miR-29 with *B7-H3* has been investigated in previous studies (see above). As mentioned earlier, hsa-miR-125a-5p regulates *B7-H5* expression in gastric cancer, but its association with *B7-H4* has not been studied. Recent findings have shown that miR-125a-5p plays a pivotal role in suppressing the classical activation of macrophages (M1-type) induced by lipopolysaccharide (LPS) stimulation, while promoting IL-4-induced expression of the alternative M2 macrophages by targeting KLF13, a transcriptional factor that is involved in T lymphocyte activation and inflammation [104]. In addition, miR-486-5p is an immunomodulatory tumor suppressor miR that has been reported to have key roles in various oncological and non-oncological disorders [105]. Although our knowledge about the role of miR-125a-5p/*B7-H4* and miR-486-5p/*B7-H4* axes in the immune pathways and the pathogenesis of various diseases is still preliminary, our in silico analysis can pave the way for further investigations.

## Figures and Tables

**Figure 1 ijms-22-02652-f001:**
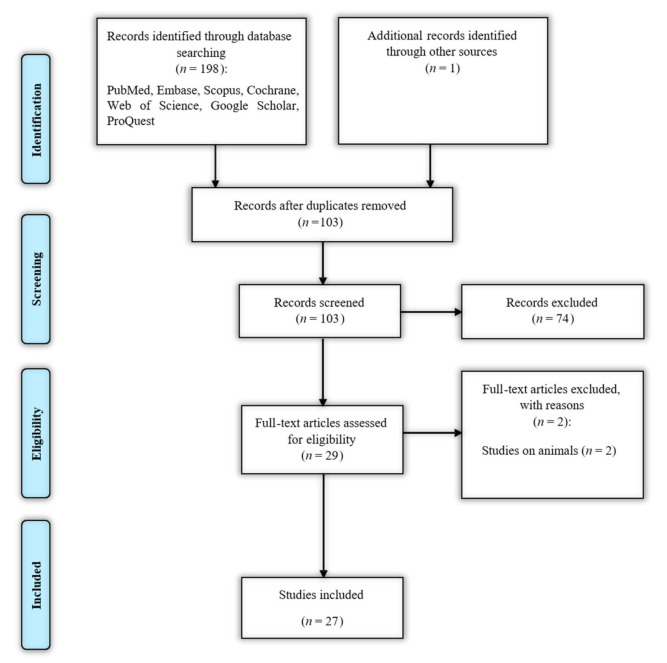
Search strategy flow chart based on the Preferred Reporting Items for Systematic Reviews and Meta-Analyses (PRISMA) flow diagram.

**Figure 2 ijms-22-02652-f002:**
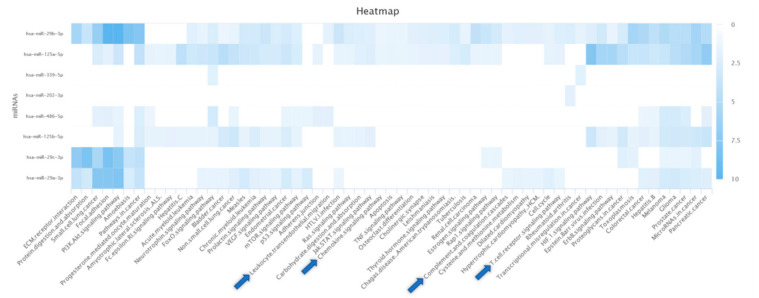
Enriched KEGG pathways of predicted miRs obtained from the miRDB database. Enriched pathways with at least two associated miRs were shown. The color of individual fields represents the −log10-transformed *p*-value of the respective enrichment results. Darker colors indicate more significant associations between an miR and the target pathway. Enriched KEGG pathways in the category of immune system are indicated by blue arrows.

**Figure 3 ijms-22-02652-f003:**
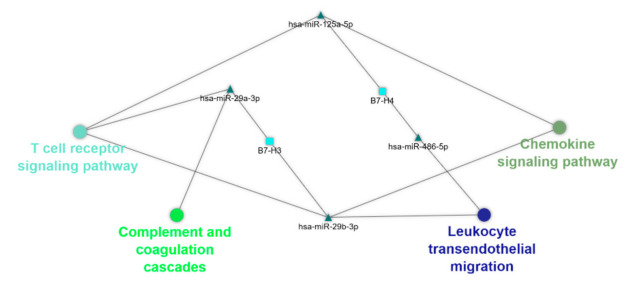
The predicted gene-miR-pathway network. Cytoscape v3.8.0 software was used for evaluating gene–miRNA interaction, and their corresponding immune-related pathways via their visualization as a network. Ellipse, triangle, and round rectangle represent pathway, miR, and gene, respectively.

**Table 1 ijms-22-02652-t001:** The characteristics of the included studies.

Regulatory Axis	Samples	Methods	Disease	Key Findings	References
miR-29/*B7-H3*	Human Burkitt lymphoma cell line Daudi, HeLa, LAN-1,NB1691, solid tumor samples (15 brain tumors, 5 hepatoblastomas, 41 neuroblastomas, 11 sarcomas, and 5 Wilms’ tumors) and 18 normal tissues	Monoclonal antibodies, whole-cell lysates, Western blot, subcellular fractionation, 8H9 antigen affinity purification, (q)RT-PCR, immunofluorescence, and luciferase reporter assay	Human cancers	Downregulation of miR-29 isoforms. Inverse regulation of *B7-H3* by miR-29. Immune escape in solid tumors.	[48]
miR-187/*B7-H3*	Blood and tissue	qRT-PCR, luciferase reporter assay, dimethylthiazol-diphenyltetrazolium bromide (MTT) assay, tumorigenicity assay, and scratch assay	Clear cell renal cell carcinoma	Downregulation of miR-187 in clear-cell renal cell carcinoma. Association with lower survival in patients. Inverse regulation of *B7-H3* by miR-187. Overexpression of miR-187 inhibits cell growth and migration.	[52]
miR-29c/*B7-H3*	Melanoma tissue and cell lines, i.e., M-1, M-101, M-111, M-12, M-14, JK-0346 Mel-B, JH-1173, and Wm266–4	IHC analysis, immunofluorescence, RT-qPCR assay, Western blotting, development of *B7-H3*-knockdown cells, development of *B7-H3*-overexpressing cells, colony formation, invasion assay, and cell migration	Cutaneous melanoma	Tumor suppressor function of miR-29c. miR-29c directly targets *B7-H3*. Overexpression of *B7-H3* increased cell migration and invasion.	[41]
miR-29c/*B7-H3*, miR-892a/*B7-H3*, miR-363/*B7-H3*, miR-940/*B7-H3*, miR-214/*B7-H3*, miR-34b/*B7-H3*, miR-665/*B7-H3*, miR-593/*B7-H3*, miR-885–3p/*B7-H3*, miR-124a/*B7-H3*, miR-326/*B7-H3*, miR-601/*B7-H3*, and miR-708/*B7-H3*	JIMT-1 cell line, KPL-4 cell line, and breast cancer tumor	Microarray screening and data analysis, immunoblotting, luciferase assays, qRT-PCR	Breast cancer	Western immunoblotting validated the 20 most effective miRs ((hsa-miR-892a, hsa-miR-380–5p, hsa-miR-125b-2, hsa-miR-363, hsa-miR-940, hsa-miR-214, hsa-miR-34b, hsa-miR-665, hsa-miR-593, hsa-miR29c, hsa-miR-555, hsa-miR-885–3p, hsa-miR-567, hsa-miR-297, hsa-miR-187–3p, hsa-miR-124a-1, hsa-miR-326, hsa-miR-601, hsa-miR-506, and hsa-miR-708) that can downregulate *B7-H3* expression in the JIMT-1 cells. Thirteen miRs directly targeted *B7-H3*. miR-29c levels were low, whereas *B7-H3* had a relatively high expression. The high miR-29c expression is associated with increased survival.	[35]
miR-29a/*B7-H3*	CL021, CL043, CL044, CL013, and tissue samples.	Sequencing, statistical analysis of differential miR expression, and qRT-PCR	Central nervous system (CNS) neuroblastoma	Low expression of miR-29a among pre-CNS primaries and CNS metastasis compared to non-CNS. *B7-H3* expression was targeted by miR-29a.	[28]
miR-124/*B7-H3*	Tumor samples and mg-63 cell line	Dual-luciferase reporter assay qRT-PCR and Western blotting analysis	Osteosarcoma (OS)	Downregulation of miR-124 in clinical OS specimens associated with advanced clinical stage and pulmonary metastasis. miR-124 directly targets *B7-H3*. Overexpression of *B7-H3* abolished the reduction of cell growth and invasion.	[42]
miR-155/CEBPB/miR-143/*B7-H3*, miR-145/*B7-H3*, miR-192/*B7-H3*, miR-378/*B7-H3* miR-155/CEBPB/miR-143/*B7-H4*	tissue samples, including cancer and adenoma tissues; Caco-2, HCT-116, LoVo, Jurkat, SW480, SW620, and CHO cell lines	Gene silencing, microRNA array, construction of miR–miR functional synergistic network, KEGG pathway enrichment analysis, qRT-PCR, immunohistochemistry, cell lysates and cell fractionations, Western blot, immunofluorescence, and ELISA and MTT assay	Colorectal cancer (CRC)	The miR-155 node was the largest in CRC. Elevated the *B7-H4* and *B7-H3* expression in adenoma tissues. miR-155 abated miR-143 expression through the transcription factor CCAAT enhancer binding protein beta (CEBPB). MiR-143 inhibited the growth of CRC cells. The lowly-expressed miR-192, miR-378, and miR-145 contributed to the *B7-H3* over-expression in colorectal cancer, consequently leading to cancer immune evasion.	[53]
miR-187/*B7-H3*	A total of 32 pairs of colorectal tumor and matched nontumor tissues and 80 CRC tissues. Human CRC cell lines SW1116, SW480, SW620, HT29, LOVO, and the normal colonic epithelial cell line NCM460.	Real-time PCR, cell proliferation assay, transwell cell migration, invasion assay, apoptosis assay, miR target prediction, Western blotting, plasmid construction, and luciferase reporter assays	Colorectal cancer	Decreased miR-187 expression shorter overall survival and relapse-free survival of patients with CRC. *B7-H3*, is negatively correlated miR-187 level in CRC cells.	[45]
miR-539/*B7-H3*	Cell lines: U87 and U251, normal human astrocytes	Cell viability assay, real-time PCR, colony formation, Western blotting, and luciferase assays	Human gliomas	Increased *B7-H3* expression and downregulation of miR-539 glioma cell lines. *B7-H3* repression by miR-539 suppresses cell proliferation in human gliomas.	[34]
miR-29c/*B7-H3*	Peripheral blood samples, human monocyte cell line THP-1	Microarray analysis of miRs, luciferase reporter assay, immunofluorescence staining, qRT-PCR, and plasma *B7-H3* detected by ELISA	Allergic asthma	The lower level of miR-29c and a higher level of plasma B7-H3 in children with asthma exacerbation. The function of miR-29c on macrophage in regulating T cell differentiation. miR-29c is correlated to its target gene *B7-H3.*	[51]
miR-29/*B7-H3*	From the CGGA dataset, we collected RNA-Seq data for 325 samples, ranging from WHO grade II to grade IV. In the TCGA dataset, RNA-Seq data were available for 669 samples.	Detection of isocitrate dehydrogenase mutation and defining immune pathways	Diffuse brain glioma	Regulation of *B7-H3* by methylation and miR-29 family at different stages, respectively. Association of *B7-H3* with higher malignancy and development and the progression of gliomas	[29]
miR-29c/*B7-H3*	Human CRC cell lines SW620 and HCT116 and human colonic epithelial cell line FHC	Cell viability assay, cell cycle assay, RT-PCR, Western blot, cycloheximide chase assay, and dual-luciferase reporter assay	Colorectal cancer	Downregulation of miR-29c in many cancer types; miR-29c directly binds to *B7-H3* mRNA and suppresses *B7-H3* expression. Astragaloside IV treatment downregulated *B7-H3* via the elevation of miR-29c.	[44]
miR-506/*B7-H3*	Bone marrow mononuclear cells were isolated from 12 de novo mantle cell lymphoma (MCL) patients with bone marrow involvement. Human MCL cell lines Maver and Z138.	qRT-PCR, Western blotting, transfection and lentivirus infection, dual-luciferase assay, cell proliferation assays, cell cycle assays, cell migration, and transwell invasion assays	Mantle cell lymphoma	Overexpression of *B7-H3* and downregulation of miR-506 in MCL patients. miR-506 inhibits the proliferation and invasion of MCL cells by targeting *B7-H3*. miR-506 induced MCL cell cycle arrest in the G0/G1 phase and repressed MCL cell migration.	[54]
miR-187/*B7-H3*	HaCaT cell line and psoriatic skin samples.	qRT-PCR, cell viability assay, Western blot analysis, RNA transfection, luciferase reporter assays, *B7-H3* overexpression (vector), histological analysis, immunohistochemistry, and cell cycle analysis	Psoriasis	Downregulation of miR-187 in the cytokine-stimulated keratinocytes compared with corresponding normal controls. Decreased miR-187 level in psoriatic skin compared with adjacent, uninvolved psoriatic skin. *B7-H3* is a direct molecular target of miR-187. Upregulation of *B7-H3* level in psoriatic skin.	[39]
MYC/miR-29/*B7-H3*	Five human medulloblastoma (MB) tumor tissues; D283 Med, D425 Med, D458 med cell line, and human umbilical vein endothelial cells (HUVECs).	In silico analysis, ELISA, human angiogenesis array, immunoblotting, gelatin zymography, F-actin staining and immunostaining, RT-PCR and RNA-Seq, immuno-paired antibody detection analysis, chromatin immunoprecipitation and DNA sequencing, fluorescence-activated cell sorting analysis, in vitro angiogenesis assay, and chick chorioallantoic membrane assay	Medulloblastoma	Association of *B7-H3* high expression with poor survival in MB patients. B7-H3 promotes angiogenesis in MB cells. miR-29 exhibits global anti-tumor functions and promotes STAT1 activation. Induces apoptosis via miR-29 in combination with MYC inhibition.	[37]
miR-1301–3p/*B7-H3*, miR-335–5p/*B7-H3* miR-28–5p/*B7-H3*	Serum samples from patients with colorectal cancer	ELISA, analysis of predicted putative miRs, and RT-PCR	Colorectal cancer	Upregulation of serum *B7-H3* expressions in CRC patients. *B7-H3* was predicted to be a target of miR-1301–3p, miR-335–5p, and miR-28–5p. Decrease serum miR-28–5p, miR-1301–3p, and miR-335–5p expressions in stage III and IV disease compared to stage I and II. These miRs are negatively related to the advanced TNM stages.	[43]
lncRNA NEAT1/miR-214/*B7-H3*	Bone marrow samples from 30 multiple myeloma patients and multiple myeloma cells, i.e., line RPMI 8226, and human monocyte cell line THP-1.	Dual-luciferase reporter assay, RNA immunoprecipitation assay, ELISA assay, Western blotting, and RT-qPCR	Multiple myeloma (MM)	Increase the level of long non-coding RNA (lncRNA) NEAT1 and *B7-H3*. Downregulation of miR-214 expression occurred in multiple myeloma tissues. lncRNA NEAT1 directly targeted miR-214 to promote M2 macrophage polarization by upregulating *B7-H3* in MM.	[30]
miR-29c/*B7-H3*/Th17	Peripheral blood and nasopharyngeal secretions samples, and human monocytic cell line THP-1	Quantitative ELISA-specific *M. pneumoniae* immunoglobulin G (IgG) and immunoglobulin M (IgM), real-time PCR for *M.* pneumoniae detection, multiple pathogen detection, examination of soluble B7-H3 and IL-17 in plasma, immunofluorescence staining, and luciferase assay	Mycoplasma pneumoniae pneumonia (MPP)	The lower level of miR-29c and a higher level of B7-H3 and IL-17 in children with MPP. *B7-H3* is the direct target of miR-29c, and miR-29c silencing or overexpression could up- or downregulate the expression of *B7-H3* in THP-1 cells.	[33]
miR-1253/*B7-H3*	All tumor samples were obtained from patients in the pediatric age group, with DAOY, D283, D341, D425, D556, D458, and HDMB03 cell lines.	PCR, cell proliferation assay, colony formation assay, cell migration and invasion, wound healing, annexin V-FITC/PI analysis, Western blotting, dual-luciferase reporter assay, target prediction in situ hybridization, immunohistochemistry, DNA methylation profiling and tumor classification, and de-methylation studies	Medulloblastoma	Deregulation of miR-1253 expression in medulloblastoma. miR-1253 inhibits mediators of cellular proliferation and promotes tumor cell apoptosis. *B7-H3* is oncogenic target of miR-1253. High expression of *B7-H3* in tumor samples. Both miR-1253 restoration and *B7-H3* silencing reduce the migratory and invasive medulloblastoma cells.	[32]
miR-145/*B7-H3*	Pleural effusion	Observational indexes and qRT-PCR	Lung cancer	The higher expression level of *B7-H3* in lung cancer. Lower expression level of miR-145 in the pleural effusion of the study group than in the control group. Relationship with lymphatic metastasis, differentiation degree, and TNM stage.	[31]
miR-199a/*B7-H3*	HeLa, C4–1, SiHa, CaSki, and C-33A	Cell proliferation assay, luciferase reporter assays, enzyme-linked immunosorbent assay, Western blot assays, qRT-PCR, cell migration, invasion assay, and immunohistochemical staining	Cervical cancer (CC)	miR-199a was expressed at lower levels in CC tissues than in adjacent normal tissues. miR-199a inhibits cell proliferation, migration and the invasion of CC cells by targeting *B7-H3*. The expression level of *B7-H3* was significantly upregulated in CC tissues.	[50]
miR-1207–5p/*B7-H4*	Whole blood samples	Selection of miR single-nucleotide polymorphisms (SNPs), SNP genotyping, selection of miR SNPs from published databases, and luciferase reporter assay	Colorectal cancer	Upregulation of miR-1207–5p in CRC tissues. miR-1207–5p can suppress the expression of B7-H4 molecule by binding with the rs13505 G-allele-specific 3’-UTR of *B7-H4* gene, which is impacted by the rs13505.	[46]
62 different miRs *	L3.6p1 cells, which were derived from a human pancreatic carcinoma	Bioinformatics analysis and miR microarray analysis	Pancreatic cancer	miRs participate in the B7-H4-mediated regulation of oncogenicity and pathogenesis of pancreatic cancer.	[38]
miR-125a-5p/*B7-H5*	Cell lines MCF10A, MDA-MB-468, MDA-MB-231, MCF7, MKN28, SNU1, MKN45, AGS, SW480, HCT116, HT29, and RKO, as well as gastric tumors	RNA extraction, expression quantification; DNA extraction, methylation analysis; short-interference-RNA experiments; BMP4 treatment; and antimiR experiments	Gastric cancer (GC)	*B7-H5* expression loss is a recurrent event in GC, caused by promoter methylation and/or miR-125a-5p overexpression, and GC-microenvironment myofibroblasts overexpress *B7-H5*. *B7-H5* expression is under the control of miR-125a-5p, as its targeted inhibition led to an overexpression of *B7-H5*.	[36]
miR-16/*B7-H5*	Colon tissue	miR microarray analysis, gene microarray analysis, qRT-PCR, miR target gene prediction, and luciferase reporter assays	Active Crohn’s disease	Upregulation of miR-16 in the inflammatory areas of the ascending colon mucosa. Inhibition of *B7-H5* expression. Immune inflammatory responses in the ascending colon. Reduction in miR-16 expression suggests the possibility of canceration of the inflammatory colon. hsa-miR-16–1 directly regulated the human *B7-H5* gene.	[47]
miR-93/*B7-H6*, miR-340/*B7-H6*, miR-195/*B7-H6*	The data were from TCGA data from 1092 patients with breast cancer, including gene expression, miR expression, and survival data. Human breast cancer cell lines, i.e., MCF7, MDA-MB-231, SK-BR-3, and human fibrocystic disease epithelium cell line MCF10A.	TCGA breast cancer transcriptome profiling analysis, sorting analysis of the most likely miRs targeting the B7 family, and qRT-PCR	Triple-negative breast cancer	Upregulation of *B7-H6* in breast cancer based on an analysis of the TCGA database. A high level of *B7-H6* suggested a worse prognosis. Bioinformatic analysis predicted that miR-93, miR-195, and miR-340 are potential regulators of the immune evasion of breast cancer cells, and they exert this function by targeting *B7-H6*.	[49]
miR-3116/*B7-H7*, miR-6870–5p/*B7-H7*	Clinical and experimental data from multiple databases, including cBioPortal, TCGA, Cistrome, TIMER, Oncomine, Kaplan–Meier, GeneXplain.	Expression profiling of *B7-H7* in human cancers, pan-cancer survival analysis, and bioinformatics analysis for understanding the regulatory mechanism of B7-H7	Clear cell carcinoma	The bioinformatic view showed that basic leucine zipper ATF-like transcription factor (BATF) in B lymphocyte and SMAD in monocytes might be responsible for the dysregulation of *B7-H7* in kidney renal clear cell carcinoma (KIRC). miR-3116 and miR-6870–5p may have a role in the regulation of *B7-H7*.	[40]

qRT-PCR: quantitative reverse transcription polymerase chain reaction, ELISA: enzyme-linked immunosorbent assay, miR: microRNA, KEGG: The Kyoto Encyclopedia of Genes and Genomes. * 62 miRs are reported in the Results section under *B7-H4*.

## Data Availability

The data presented in this study are available in this review.

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
