# Peer review of "The Regulatory Cross-Talk between microRNAs and Novel Members of the B7 Family in Human Diseases: A Scoping Review"

_ijms, 2021, doi:10.3390/ijms22052652_

Round 1

Reviewer 1 Report

This is an interesting study suggesting a scoping review related to the B7 family.
-There are several typos in the text. Please work on it.
-Please check the sort of references that are not cited well
-The quality of Figure 2 and Figure 3 is low.

-There are some grammatical mistakes in the text, ask a native speaker to read it and work on it.

Author Response

Reviewer #1:

This is an interesting study suggesting a scoping review related to the B7 family.

  1. There are several typos in the text. Please work on it.

We are thankful for this comment. Based on the reviewer's comment, the manuscript was edited carefully.

  1. Please check the sort of references that are not cited well

We are appreciative of the careful consideration of the reviewer. Based on the reviewer's comment, we have corrected the reference list.

  1. The quality of Figure 2 and Figure 3 is low.

We want to extend our appreciation for the detailed consideration given to our manuscript. According to the reviewer's comment, the quality of the abovementioned figures was enhanced to better present information.

  1. There are some grammatical mistakes in the text, ask a native speaker to read it and work on it.

We are thankful for this comment. Based on the reviewer's comment, the manuscript was edited carefully by a native English speaker.

Reviewer 2 Report

In their manuscript, Ahangar and colleagues review the existing literature on microRNA regulation of B7 family immune checkpoint proteins. The authors also perform a bioinformatics analysis of experimentally validated microRNAs regulating these proteins identified via search in mirWalk database and conduct a KEGG pathway enrichment analysis to find possible pathways these microRNAs and their targets may be involved in. While this manuscript may be potentially interesting for specialists in the field, there are several concerns which, in my view, should be addressed before it can be considered for publication.

Below, please, find the detailed points.

The manuscript should be proofread; annoying absence of spaces between the words throughout the text makes reading the text extremely irritating.

The text on figure 2 and especially on figure 3 is absolutely impossible to read, even at high magnification in digital version, not to say about the print version of the figures. As such, these figures are completely useless. The authors should split the figure 2 in different panels, increasing font size on the drawings. It would be more informative, if the microRNAs which are supported by reviewed literature would have also been listed and labeled on the figure. For the figure 3, I do not see the point of showing it at all in this present form. The text is absolutely unreadable, there is no color legend, individual microRNAs are indistinguishable. What do the authors want to show by this? The authors should explain these results more in the results section. Also in the discussion section, the discussion of bioinformatics results should be expanded. The authors briefly discuss only type 2 diabetes, what about other pathways? The identified pathways are not directly related to immune response, but, surprisingly, the authors do not comment on this at all. As such, I do not see any scientific value in these results presented in the current form, unless they are properly depicted, carefully described and adequately discussed.

In the introduction (page 2), the authors are wrong in describing mechanism of action of microRNAs. miRs do not cleave their target mRNAs by themselves; they serve as essential components guiding multiprotein RISC complex to corresponding targets. The authors are strongly advised to familiarize themselves with and cite a recent review on microRNA biogenesis and functions by David Bartel (https://pubmed.ncbi.nlm.nih.gov/29570994/) and correct their manuscript accordingly.

In the literature analysis (page 4), what was the reason to exclude 47 articles after the title and abstract screening?

The table is arguably the most valuable part of this manuscript. However, the data in the table should be rearranged to focus on published microRNAs and their targets. As such, the column ”Regulatory axis” should be the first (sorted according to particular reported miR-mRNA interaction), followed by the information on samples, methods, disease, key findings and the references. In the current form, the first four columns are redundant, as the information on corresponding publication is given in the last column.

I do not really understand the following sentence on page 8: “A study has demonstrated no correlation between  miR-29c  and  soluble  B7-H3  in  children  withMycoplasma  pneumoniae pneumonia;however, there has been a negative correlation between B7-H3 and miR-29c [28].” – so is there a correlation or not?

Minor points:

The last sentence of the introduction part (page 2): “Results of this review might pave the way for introducing new paradigms for miR-based immunotherapy.” is extremely vague. How do the authors envision the application of these results in reality?

MicroRNA identifiers in last paragraph on page 10 should read “hsa-“, not “has-“.

Author Response

Reviewer #2:

In their manuscript, Ahangar and colleagues review the existing literature on microRNA regulation of B7 family immune checkpoint proteins. The authors also perform a bioinformatics analysis of experimentally validated microRNAs regulating these proteins identified via search in mirWalk database and conduct a KEGG pathway enrichment analysis to find possible pathways these microRNAs and their targets may be involved in. While this manuscript may be potentially interesting for specialists in the field, there are several concerns which, in my view, should be addressed before it can be considered for publication. Below, please, find the detailed points.

  1. The manuscript should be proofread; annoying absence of spaces between the words throughout the text makes reading the text extremely irritating.

We are thankful for this comment. The reviewer has raised a valid concern. Indeed, we have also noticed the unintentional mistake after the submission of the manuscript. We hope that our manuscript will be reached to the reviewer as we upload it. Furthermore, the manuscript was edited and proofed carefully, according to the reviewer's comment.

  1. The text on figure 2 and especially on figure 3 is absolutely impossible to read, even at high magnification in digital version, not to say about the print version of the figures. As such, these figures are completely useless. The authors should split the figure 2 in different panels, increasing font size on the drawings. It would be more informative, if the microRNAs which are supported by reviewed literature would have also been listed and labeled on the figure. For the figure 3, I do not see the point of showing it at all in this present form. The text is absolutely unreadable, there is no color legend, individual microRNAs are indistinguishable. What do the authors want to show by this?

We are grateful for this valuable comment. We improved the quality of the abovementioned figures to present the information better.

  1. The authors should explain these results more in the results section. Also in the discussion section, the discussion of bioinformatics results should be expanded. The authors briefly discuss only type 2 diabetes, what about other pathways? The identified pathways are not directly related to immune response, but, surprisingly, the authors do not comment on this at all. As such, I do not see any scientific value in these results presented in the current form, unless they are properly depicted, carefully described and adequately discussed.

We want to extend our appreciation for such a meticulous consideration given to our manuscript. The reviewer has raised a valid concern; thus, we edited the manuscript accordingly.

In the current version of this manuscript, we used the miRDB database to predict other possible B7/miRNA regulatory axes. After obtaining the human functional miRs with scores over 80, we performed the KEGG pathway enrichment analysis focusing on the "Immune system." Four regulatory axes, which were annotated for KEGG pathways in the category immune system, were identified. Two of them were supported by reviewed literature. We discussed the other two and provided the rationale for further investigations in the future. Moreover, the figures were corrected according to the new bioinformatics approach with enhanced quality.

  1. In the introduction (page 2), the authors are wrong in describing mechanism of action of microRNAs. miRs do not cleave their target mRNAs by themselves; they serve as essential components guiding multiprotein RISC complex to corresponding targets. The authors are strongly advised to familiarize themselves with and cite a recent review on microRNA biogenesis and functions by David Bartel (https://pubmed.ncbi.nlm.nih.gov/29570994/) and correct their manuscript accordingly.

We are grateful for this valuable comment. We have corrected the aforementioned statement by focusing on the mentioned paper to present the manuscript in a more accurate manner.

  1. In the literature analysis (page 4), what was the reason to exclude 74 articles after the title and abstract screening?

We are thankful for this comment. Indeed, seventy-four records were excluded because their data could not answer our research question. We revised the manuscript accordingly.  

  1. The table is arguably the most valuable part of this manuscript. However, the data in the table should be rearranged to focus on published microRNAs and their targets. As such, the column "Regulatory axis" should be the first (sorted according to particular reported miR-mRNA interaction), followed by the information on samples, methods, disease, key findings and the references. In the current form, the first four columns are redundant, as the information on corresponding publication is given in the last column.

We are appreciative of this valuable comment. We revised the manuscript accordingly.

  1. I do not really understand the following sentence on page 8: "A study has demonstrated no correlation between miR-29c and soluble B7-H3 in children with Mycoplasma pneumoniae pneumonia; however, there has been a negative correlation between B7-H3 and miR-29c [28]." – so is there a correlation or not?

We are thankful for the detailed consideration given to our manuscript. We revised the abovementioned statement according to the reviewer's comment.

Minor points:

  1. The last sentence of the introduction part (page 2): "Results of this review might pave the way for introducing new paradigms for miR-based immunotherapy." is extremely vague. How do the authors envision the application of these results in reality?

We are appreciative of the careful consideration of the reviewer. Based on the reviewer's comment, we have revised the aforementioned statement accordingly.

  1. MicroRNA identifiers in last paragraph on page 10 should read "hsa-", not "has-".

We are thankful for this comment. Based on the reviewer's comment, we have edited that statement accordingly.